# Improving Span Representation by Efficient Span-Level Attention

**Pengyu Ji, Songlin Yang, Kewei Tu**[*]
School of Information Science and Technology, ShanghaiTech University
Shanghai Engineering Research Center of Intelligent Vision and Imaging
{jipy2023,yangsl,tukw}@shanghaitech.edu.cn

## Abstract

High-quality span representations are crucial to natural language processing tasks involving span prediction and classification. Most existing methods derive a span representation by aggregation of token representations within the span. In contrast, we aim to improve span representations by considering span-span interactions as well as more comprehensive span-token interactions. Specifically, we introduce layers of span-level attention on top of a normal token-level transformer encoder. Given that attention between all span pairs results in $O(n^4)$ complexity ($n$ being the sentence length) and not all span interactions are intuitively meaningful, we restrict the range of spans that a given span could attend to, thereby reducing overall complexity to $O(n^3)$. We conduct experiments on various span-related tasks and show superior performance of our model surpassing baseline models. Our code is publicly available at https://github.com/jipy0222/Span-Level-Attention.

## 1 Introduction

Many natural language processing tasks involve spans, making it crucial to construct high-quality span representations. In named entity recognition, spans are detected and typed with different labels (Yuan et al., 2022; Zhu et al., 2022); in coreference resolution, mention spans are located and grouped (Lee et al., 2017, 2018; Gandhi et al., 2021; Liu et al., 2022); in constituency parsing, spans are assigned scores for constituent labels, based on which a parse tree structure is derived (Stern et al., 2017; Kitaev and Klein, 2018; Kitaev et al., 2019).

Most existing methods compute span representations by shallowly aggregating token representations. They either pool over tokens within the span (Shen et al., 2021; Hashimoto et al., 2017; Conneau et al., 2017), or concatenate the starting and ending tokens (Ouchi et al., 2018; Zhong and Chen,

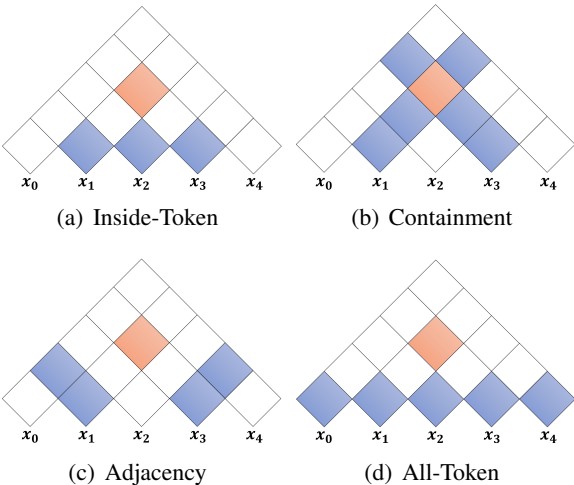

Figure 1: Diagrams for four attention patterns. Each cell represents a span, e.g., the orange cell in each diagram represents the span consisting of tokens from $x_1$ to $x_3$. Orange cells represent target spans and blue cells represent spans they can attend to.

2021). The limitation of these methods lies in: (i) Span representations are dominated by a subset of tokens, resulting in a potential lack of crucial information. (ii) Intuitively, span interactions should play an important role in span encoding. For example, meanings of spans, especially constituents, can be composed from their sub-spans and disambiguated by their neighbouring spans. However, such span interactions are completely ignored in these methods.

Inspired by the utilization of self-attention in Transformer (Vaswani et al., 2017), we introduce span-level self-attention to capture span interactions and improve span representations. However, computing attention scores for all span pairs leads to $O(n^4)$ complexity ($n$ for sequence length). In addition, not all span interactions are intuitively meaningful. Therefore, we design four different span-level patterns to restrict the range of spans that a given span could attend to: Inside-Token,

---

[*]Corresponding author.

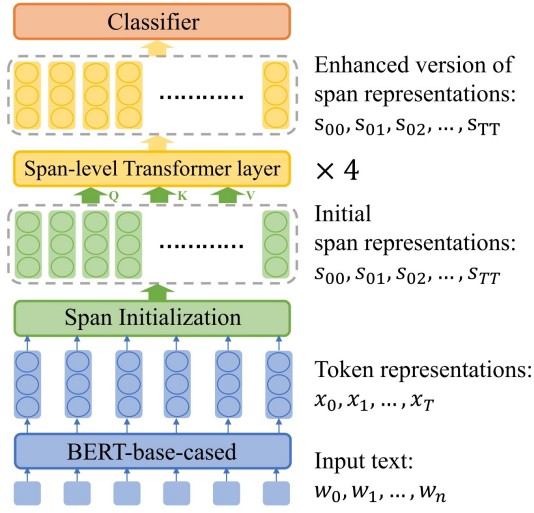

Figure 2: Architecture of our model.

Containment, Adjacency and All-Token (Fig. 1). Each of them allows only $O(n)$ spans for attention, reducing the overall complexity to $O(n^3)$.

Many existing studies also aim at improving span representations. Yuan et al. (2022) utilize entity labels to define specific span representations for nested NER. Zhou et al. (2022) use syntactic parse trees to enhance span encoding. Zhu et al. (2022) improve span representations by stacking multiple span-token attention layers. Wang et al. (2022) introduce intra-span attention to enhance span representations by computing attention between each given span and all other spans. Compared to existing works, our method offers unique advantages: (i) We design span representations for a wide range of tasks without relying on external information such as labels and parse trees. (ii) We lay more emphasis on span interactions by incorporating span-level attention. (iii) We design span-level attention patterns to capture meaningful span interactions and reduce the overall complexity to an acceptable level, thereby ensuring both effectiveness and efficiency.

## 2 Method

Fig. 2 illustrates the architecture of our model, which we describe from bottom up.

**Token representations.** Given a sentence $w = w_0, w_1, \ldots, w_n$, we pass it through BERT (Devlin et al., 2019) to do tokenization and obtain contextualized token representations $c = c_0, c_1, \ldots, c_T$ by taking a weighted average of the outputs from all layers. We then feed them into a linear pro-

jection layer to obtain final token representations $x = x_0, x_1, \ldots, x_T$.

**Initial span representations.** We follow Toshniwal et al. (2020) to initialize span representations from contextualized token representations. During pilot experiments[1], we observe that among the five pooling methods (max pooling, average pooling, attention pooling, endpoint, diff-sum), max pooling performs the best. Therefore, we choose max pooling as the default initialization method. Specifically, given a span $\langle i, j \rangle$ and the corresponding token representations $\{x_i, \ldots, x_j\}$ within the span, the initial span representation $s_{ij}$ is computed by selecting the maximum value over each dimension of the token representations.

**Span-level attention.** We enumerate all the spans and input their representations to a Transformer encoder with span-level attention. Note that computing attention scores for all span pairs leads to $O(n^4)$ time and memory complexity because self-attention has a quadratic complexity and there are a total of $O(n^2)$ spans. To reduce the complexity as well as encourage more meaningful span interactions, we design different attention patterns to restrict the range of spans that a given span could attend to (Fig. 1). We use $rel(\langle i, j \rangle)$ to denote the set of spans that span $\langle i, j \rangle$ can attend to.

**Inside-Token** Each span attends to tokens within this span. This pattern maintains the connection between spans and their internal tokens.

$$rel(\langle i, j \rangle) = \{\langle k, k \rangle | k = i, \ldots, j\}$$

**Containment** Given a span, its super-spans and sub-spans may provide meaningful information of the span. However, the total number of super-spans and sub-spans of a given span is $O(n^2)$. Considering the importance of starting and ending positions in span encoding, we propose that each span attends to spans that share the same starting or ending position. This pattern takes into account the containment relationship as well as the starting and ending positions of spans, while reducing the number of spans to $O(n)$.

$$rel(\langle i, j \rangle) = \{\langle i, k \rangle | k = i, \ldots, T\} \cup \{\langle k, j \rangle | k = 0, \ldots, j\}$$

---

[1] Results can be found in Table 7 in Appendix B

| | Pattern | | | | Task | | | | | | |
|---|---|---|---|---|---|---|---|---|---|---|---|
| | (a) | (b) | (c) | (d) | NEL | REF | SRC | CTL | MED | CTD | Avg. |
| i | Max pooling | | | | 95.61 | 95.58 | 92.85 | 98.00 | 97.98 | 98.19 | 96.37 |
| ii | Best pooling | | | | 95.78 | 95.78 | 92.86 | 98.00 | 98.10 | 98.24 | 96.46 |
| iii | Token-level | | | | 95.73 | 95.76 | 93.32 | 98.34 | 98.04 | 98.61 | 96.63 |
| iv | Fully-connected | | | | 95.77 | 95.63 | 93.11 | 98.36 | 98.12 | 98.69 | 96.61 |
| 1. | ✓ | ✗ | ✗ | ✗ | 95.57 | 95.80 | **93.62** | 98.39 | 98.10 | 98.53 | 96.67 |
| 2. | ✗ | ✓ | ✗ | ✗ | 95.85 | 95.93 | 93.46 | 98.45 | 98.19 | 98.79 | 96.78 |
| 3. | ✗ | ✗ | ✓ | ✗ | 95.79 | 95.63 | 93.46 | 98.47 | 98.24 | 98.79 | 96.73 |
| 4. | ✗ | ✗ | ✗ | ✓ | 95.85 | **96.06** | 93.31 | 98.35 | 98.05 | 98.62 | 96.71 |
| 5. | ✓ | ✓ | ✗ | ✗ | 95.94 | 96.02 | 93.52 | 98.45 | 98.23 | 98.83 | **96.83** |
| 6. | ✓ | ✗ | ✓ | ✗ | 95.78 | 95.78 | 93.50 | **98.50** | **98.28** | **98.83** | 96.78 |
| 7. | ✓ | ✗ | ✗ | ✓ | 95.73 | 95.98 | 93.33 | 98.36 | 98.02 | 98.56 | 96.66 |
| 8. | ✗ | ✓ | ✓ | ✗ | 95.76 | 95.92 | 93.50 | 98.47 | 98.19 | 98.79 | 96.77 |
| 9. | ✗ | ✓ | ✗ | ✓ | **96.08** | 95.77 | 93.35 | 98.43 | 98.14 | 98.77 | 96.76 |
| 10. | ✗ | ✗ | ✓ | ✓ | 95.95 | 95.80 | 93.31 | 98.44 | 98.23 | 98.80 | 96.75 |
| 11. | ✓ | ✓ | ✓ | ✗ | 95.85 | 95.68 | 93.52 | 98.48 | 98.22 | 98.81 | 96.76 |
| 12. | ✓ | ✓ | ✗ | ✓ | 95.87 | 95.89 | 93.36 | 98.44 | 98.18 | 98.80 | 96.76 |
| 13. | ✓ | ✗ | ✓ | ✓ | 95.79 | 95.73 | 93.29 | 98.45 | 98.25 | 98.80 | 96.72 |
| 14. | ✗ | ✓ | ✓ | ✓ | 95.80 | 95.81 | 93.33 | 98.41 | 98.24 | 98.77 | 96.73 |
| 15. | ✓ | ✓ | ✓ | ✓ | 95.82 | 95.80 | 93.38 | 98.43 | 98.20 | 98.76 | 96.73 |

Table 1: Averaged F1 scores for 6 probing tasks with baselines and different pattern combinations. (a): Inside-Token, (b): Containment, (c): Adjacency, (d): All-Token.

**Adjacency** Each span attends to spans that share only the starting or ending positions of the span. Intuitively, adjacent spans often have strong correlations.

$$rel(\langle i,j \rangle) = \{\langle j,k \rangle | k = j, \ldots, T\} \cup$$
$$\{\langle k,i \rangle | k = 0, \ldots, i\}$$

**All-Token** Each span attends to all tokens in the input text. This pattern enables the acquisition of token information beyond span boundaries.

$$rel(\langle i,j \rangle) = \{\langle k,k \rangle | k = 0, \ldots, T\}$$

It is worth noting that all four patterns ensure that the number of spans each span can attend to is $O(n)$, which reduces the overall complexity to $O(n^3)$. Moreover, we can combine these four patterns arbitrarily to form new patterns when facing different scenarios.

**Inference and Training.** After span-level attention, we obtain an enhanced version of $s_{ij}$ for each span. For single span tasks, we feed a span representation into a two-layer MLP classifier. For tasks involving two spans, we concatenate the two span representations and feed them into the MLP classifier. The classifier maps the input into a $q$-dimensional vector, where $q$ is the size of the label set (including NoneType if necessary). We directly utilize the loss function of downstream tasks to train the model, such as the commonly used binary cross-entropy loss and cross-entropy loss in multi-class classification tasks.

## 3 Experiment

### 3.1 Setup

We use BERT-base-cased to obtain contextualized token representations and keep it frozen when conducting probing tasks. We stack 4 Transformer encoder layers and set the number of heads in multi-head attention to 4 to do span-level attention. Dataset details and other hyper-parameters can be found in Table 5 and Table 6 in Appendix A. We conduct all experiments on a single 24GB NVIDIA TITAN RTX and report the micro-averaged F1-scores. All results are averaged over three runs with different random seeds.

### 3.2 Probing tasks results

We conduct 6 probing tasks: named entity labeling (NEL), coreference arc prediction (REF), semantic role classification (SRC), constituent labeling (CTL), mention detection (MED) and constituent detection (CTD), following Toshniwal et al. (2020). In these 6 tasks, we only need to do classification or prediction on given spans.

Table 1 shows probing tasks results. We pose (i) max pooling, (ii) best performing pooling among five pooling methods mentioned in section 2, (iii) max pooling after four additional layers of normal token-level attention, and (iv) fully-connected span-level attention (i.e., the $O(n^4)$ full span-level attention without restriction) as four baselines[2]. Overall,

---

[2] Stacking four layers of fully-connected span-level attention can result in the out-of-memory issue when doing ex-

fully-connected span-level attention shows good performance compared to pooling methods, validating the effectiveness of span-level attention. Furthermore, applying different attention patterns or pattern combinations not only reduces computational complexity, but also significantly improves performance. This suggests that our proposed attention patterns effectively capture more meaningful span interactions than fully-connected span-level attention without restrictions. Our method also outperforms token-level attention with additional layers, suggesting that the improvement in performance is not merely due to having more parameters.

For specific tasks, the optimal attention patterns vary. For tasks that place more emphasis on structures, such as CTL, CTD and detection task MED, attention patterns inspired by structural span interactions (Containment, Adjacency) show better performance. The same applies to pattern combinations involving them. This makes sense because grammatical structures are closely related to the structural span interactions within a sentence. For tasks that prioritize textual content, such as REF, the All-Token attention pattern performs better due to its attention to the entire input text. In SRC, we speculate that the Inside-Token pattern helps us focus specifically on the prefixes or suffixes generated by tokenization, thus improving performance related to semantic roles. In NEL, a combination of the Containment and All-Token patterns strikes a balance between structure and semantics, leading to good performance.

In general, as shown in Table 8 in Appendix B, our method consistently outperforms the baseline models in all 6 tasks. Moreover, the best performing attention pattern is the combination of the Inside-Token and Containment patterns. This pattern combination, due to its consideration of both semantics and structure, is a reliable choice across different tasks.

### 3.3 Nested NER results

We conduct nested named entity recognition (nested NER) on the ACE2004[3] and ACE2005[4] datasets (Doddington et al., 2004).

As Table 2 shows, significant improvements are

---

periments on 24GB NVIDIA TITAN RTX due to its $O(n^4)$ complexity, so we do experiments on 48GB NVIDIA A40 instead.

[3] https://catalog.ldc.upenn.edu/LDC2005T09
[4] https://catalog.ldc.upenn.edu/LDC2006T06

| | Encoders&Datasets | | | | | |
| | BERT-frozen | | | BERT-finetune | | |
| i | ACE04 | ACE05 | Avg. | ACE04 | ACE05 | Avg. |
|---|---|---|---|---|---|---|
| i | 75.54 | 76.90 | 76.22 | 84.12 | 82.56 | 83.34 |
| 1. | 79.18 | 79.38 | 79.28 | 83.92 | 82.94 | 83.43 |
| 2. | 80.00 | 80.01 | 80.01 | 84.23 | 83.50 | 83.86 |
| 3. | 80.61 | 79.97 | 80.29 | 84.38 | 83.52 | 83.95 |
| 4. | 77.39 | 78.97 | 78.18 | 83.71 | 83.31 | 83.51 |
| 5. | 80.37 | **80.57** | 80.47 | 84.33 | 83.35 | 83.84 |
| 6. | **81.31** | 80.36 | **80.83** | 84.54 | 83.63 | 84.08 |
| 7. | 77.69 | 78.58 | 78.14 | 84.54 | 82.72 | 83.63 |
| 8. | 79.65 | 79.67 | 79.66 | 84.21 | 83.20 | 83.70 |
| 9. | 79.54 | 79.90 | 79.72 | 84.25 | 83.60 | 83.93 |
| 10. | 79.49 | 79.09 | 79.29 | 83.83 | 83.17 | 83.50 |
| 11. | 80.23 | 80.07 | 80.15 | 84.57 | 83.54 | 84.05 |
| 12. | 79.49 | 80.23 | 79.86 | 84.32 | 83.49 | 83.90 |
| 13. | 79.14 | 79.11 | 79.12 | 84.58 | **83.70** | 84.14 |
| 14. | 79.66 | 80.38 | 80.02 | **84.97** | 83.56 | **84.26** |
| 15. | 78.86 | 79.80 | 79.33 | 84.36 | 83.50 | 83.93 |

Table 2: Averaged F1 scores for nested NER with baseline and different pattern combinations. We use the same index as Tab. 1 in the first column to represent the same model.

observed with span-level attention compared to max pooling when freezing BERT. Fine-tuning BERT leads to further enhancements in overall performance. We speculate that combining span-level attention with stronger pretrained language models and carefully-designed decoders will yield even better results. Specifically, the combination of Inside-Token and Containment/Adjacency performs well when using a frozen BERT, which aligns with our observations from probing tasks conducted under similar settings. When BERT is fine-tuned, token representations capture more comprehensive contextual information, allowing the All-Token pattern to be included in the optimal combination. Table 9 in Appendix B also shows that the improvements brought about by our method are consistent.

### 3.4 SpanBERT backbone

We also conduct experiments on the REF and SRC tasks with SpanBERT being used as the backbone to analyse the generalizability of our proposed method. As Table 3 shows, span-level attention still brings performance gain after changing the backbone to SpanBERT, no matter fine-tuned or not. It further demonstrates the generalizability of our method. Note that fine-tuned SpanBERT with max pooling is a widely favored choice for span-related tasks, and our results show that applying span-level attention to this backbone can still bring performance improvement. More detailed results can be found in Table 10 in Appendix B.

| | SpanBERT-frozen | | | SpanBERT-finetune | | |
|---|---|---|---|---|---|---|
| | REF | SRC | Avg. | REF | SRC | Avg. |
| i | 95.62 | 92.88 | 94.25 | 96.52 | 93.66 | 95.09 |
| 1. | 95.90 | 93.40 | 94.65 | 96.65 | 93.77 | 95.21 |
| 2. | 95.83 | 93.39 | 94.61 | 96.67 | 93.86 | 95.27 |
| 3. | 95.73 | 93.43 | 94.58 | 96.52 | 93.81 | 95.17 |
| 4. | 95.93 | 93.22 | 94.58 | 96.73 | 93.77 | 95.25 |
| 5. | 95.86 | 93.45 | 94.66 | 96.73 | 93.78 | 95.26 |
| 6. | 95.87 | **93.51** | **94.69** | 96.71 | **93.87** | 95.29 |
| 7. | **96.01** | 93.12 | 94.57 | 96.62 | 93.80 | 95.21 |
| 8. | 95.62 | 93.47 | 94.55 | 96.59 | 93.84 | 95.22 |
| 9. | 95.94 | 93.25 | 94.60 | **96.83** | 93.79 | 95.31 |
| 10. | 95.92 | 93.19 | 94.56 | 96.69 | 93.81 | 95.25 |
| 11. | 95.97 | 93.41 | **94.69** | 96.60 | 93.85 | 95.23 |
| 12. | 96.00 | 93.28 | 94.64 | 96.74 | 93.78 | 95.26 |
| 13. | 95.93 | 93.21 | 94.57 | 96.80 | **93.87** | **95.34** |
| 14. | 95.78 | 93.16 | 94.47 | 96.80 | 93.84 | 95.32 |
| 15. | 95.87 | 93.25 | 94.56 | 96.59 | 93.75 | 95.17 |

Table 3: Averaged F1 scores for REF and SRC with baseline and different pattern combinations when Span-BERT is used as the backbone. We use the same index as Tab. 1 in the first column to represent the same model.

| Attention Layer | Avg. F1 improvement |
|---|---|
| # layer = 2 | 6.24 |
| # layer = 4 | **6.30** |
| # layer = 6 | 5.98 |

Table 4: Analysis on span-level attention layer.

### 3.5 Analysis

We conduct an analysis on the effect of the number of span-level attention layers. We select three tasks, REF, SRC and CTL, to cover both semantic and structural tasks. To compare the results, we calculate the overall improvements of different attention patterns compared to the max pooling baseline and average them across the three tasks. As Table 4 shows, stacking 4 layers slightly outperforms the other two options.

### 4 Conclusion

We propose to use span-level attention to improve span representations. In order to reduce the $O(n^4)$ complexity and encourage more meaningful span interactions, we incorporate different attention patterns to limit the scope of spans that a particular span can attend to. Experiments on various tasks validate the efficiency and effectiveness of our method.

### Limitations

We conduct an empirical study with extensive experiments to validate the effectiveness of our pro-posed method and attempt to derive further observations from the experiments. However, there is a lack of solid theoretical explanations and insights for these observations.

Moreover, It can be time-consuming to try different pattern combinations and pick the optimal one when encountering new tasks. To enhance efficiency, one possible approach is to propose an automated attention pattern combiner based on reinforcement learning, which can serve as an important component of the entire model.

### Acknowledgements

This work was supported by the National Natural Science Foundation of China (61976139).

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

# A   Implementation details

We provide dataset statistics and hyper-parameters summary in Table 5 and 6. We filter out sentences with length exceeding 40 in probing task datasets and 100 in nested NER datasets. Moreover, we evaluate the model on development set every 500 steps while training. If no improvement is observed in the previous 5 evaluations, the learning rate is reduced by a factor of 2. The gradient accumulation step is set to 8 for SRC, CTL, CTD and 4 for NEL, REF, MED and nested NER.

| Task | $|\mathcal{L}|$ | #Instances(Train/Val./Test) |
|---|---|---|
| **NEL** | 18 | 103K / 16K / 10K |
| **REF** | 2 | 161K / 19K / 21K |
| **SRC** | 62 | 510K / 71K / 52K |
| **CTL** | 30 | 1.6M / 215K / 158K |
| **MED** | 2 | 718K / 86K / 90K |
| **CTD** | 2 | 2.6M / 354K / 259K |
| **NER(ACE04)** | 7 | 22K / 3K / 3K |
| **NER(ACE05)** | 7 | 24K / 3K / 3K |

Table 5: Dataset statistics.

| **Architecture hyper-parameters** | |
|---|---|
| Span representation dimension | 256 |
| Span-level attention head | 4 |
| Span-level attention layer | 4 |
| Span-level attention FFN dimension | 1024 |
| Span-level attention dropout | 0.1 |
| Span-level attention layernorm eps | 1e-5 |
| Classifier hidden dimension | 256 |
| Classifier dropout | 0.2 |
| Classifier layernorm eps | 1e-5 |
| **Training-related hyper-parameters** | |
| Training epoch | 20 |
| Batch size | 16 |
| BERT learning rate | 5e-5 |
| Span-level attention learning rate | 2e-4 |
| Other learning rate | 5e-4 |
| Optimizer | Adam |

Table 6: Summary of hyper-parameters.

# B  Detailed results

We provide detailed experiment results in this section. Table 7, 8, 9 shows averaged F1 scores along with standard deviations in pilot experiments, probing tasks and nested NER. Table 10 shows REF and SRC results when SpanBERT is used as backbone.

| | NEL | REF | SRC | CTL | MED | CTD | Avg. |
|---|---|---|---|---|---|---|---|
| max pooling | 95.61±0.09 | 95.58±0.09 | 92.85±0.09 | **98.00±0.01** | 97.98±0.05 | 98.19±0.02 | **96.37±0.02** |
| average pooling | 95.47±0.06 | 95.21±0.08 | 92.00±0.04 | 95.84±0.12 | 97.76±0.01 | 96.75±0.14 | 95.50±0.04 |
| attention pooling | **95.78±0.09** | **95.78±0.05** | 92.86±0.10 | 97.34±0.07 | 97.87±0.01 | 97.37±0.11 | 96.17±0.01 |
| endpoint | 95.56±0.05 | 95.30±0.09 | **92.86±0.02** | 97.77±0.02 | 98.09±0.01 | **98.24±0.01** | 96.30±0.02 |
| diff-sum | 95.47±0.09 | 95.39±0.05 | 92.81±0.06 | 97.76±0.02 | **98.10±0.01** | 98.24±0.03 | 96.29±0.02 |

Table 7: Detailed results for pilot experiments.

| | Pattern | | | | Task | | | | | | |
|---|---|---|---|---|---|---|---|---|---|---|---|
| | (a) | (b) | (c) | (d) | NEL | REF | SRC | CTL | MED | CTD | Avg. |
| i | Max pooling | | | | 95.61±0.09 | 95.58±0.09 | 92.85±0.09 | 98.00±0.01 | 97.98±0.05 | 98.19±0.02 | 96.37±0.02 |
| ii | Best pooling | | | | 95.78±0.09 | 95.78±0.05 | 92.86±0.10 | 98.00±0.01 | 98.10±0.01 | 98.24±0.01 | 96.46±0.01 |
| iii | Token-level | | | | 95.73±0.06 | 95.76±0.05 | 93.32±0.02 | 98.34±0.02 | 98.04±0.02 | 98.61±0.02 | 96.63±0.02 |
| iv | Fully-connected | | | | 95.77±0.15 | 95.63±0.03 | 93.11±0.02 | 98.36±0.01 | 98.12±0.04 | 98.69±0.01 | 96.61±0.03 |
| 1. | ✓ | ✗ | ✗ | ✗ | 95.57±0.11 | 95.80±0.09 | **93.62±0.02** | 98.39±0.01 | 98.10±0.02 | 98.53±0.01 | 96.67±0.02 |
| 2. | ✗ | ✓ | ✗ | ✗ | 95.85±0.02 | 95.93±0.01 | 93.46±0.03 | 98.45±0.03 | 98.19±0.01 | 98.79±0.02 | 96.78±0.01 |
| 3. | ✗ | ✗ | ✓ | ✗ | 95.79±0.06 | 95.63±0.06 | 93.46±0.03 | 98.47±0.02 | 98.24±0.02 | 98.79±0.03 | 96.73±0.02 |
| 4. | ✗ | ✗ | ✗ | ✓ | 95.85±0.03 | **96.06±0.02** | 93.31±0.04 | 98.35±0.01 | 98.05±0.04 | 98.62±0.01 | 96.71±0.01 |
| 5. | ✓ | ✓ | ✗ | ✗ | 95.94±0.12 | 96.02±0.12 | 93.52±0.01 | 98.45±0.05 | 98.23±0.02 | 98.83±0.01 | **96.83±0.02** |
| 6. | ✓ | ✗ | ✓ | ✗ | 95.78±0.07 | 95.78±0.13 | 93.50±0.04 | **98.50±0.01** | **98.28±0.04** | **98.83±0.01** | 96.78±0.04 |
| 7. | ✓ | ✗ | ✗ | ✓ | 95.73±0.13 | 95.98±0.05 | 93.33±0.04 | 98.36±0.03 | 98.02±0.03 | 98.56±0.02 | 96.66±0.02 |
| 8. | ✗ | ✓ | ✓ | ✗ | 95.76±0.09 | 95.92±0.10 | 93.50±0.03 | 98.47±0.01 | 98.19±0.03 | 98.79±0.03 | 96.77±0.03 |
| 9. | ✗ | ✓ | ✗ | ✓ | **96.08±0.09** | 95.77±0.02 | 93.35±0.01 | 98.43±0.01 | 98.14±0.04 | 98.77±0.03 | 96.76±0.02 |
| 10. | ✗ | ✗ | ✓ | ✓ | 95.95±0.12 | 95.80±0.13 | 93.31±0.07 | 98.44±0.02 | 98.23±0.04 | 98.80±0.01 | 96.75±0.04 |
| 11. | ✓ | ✓ | ✓ | ✗ | 95.85±0.07 | 95.68±0.02 | 93.52±0.04 | 98.48±0.01 | 98.22±0.02 | 98.81±0.04 | 96.76±0.03 |
| 12. | ✓ | ✓ | ✗ | ✓ | 95.87±0.06 | 95.89±0.05 | 93.36±0.05 | 98.44±0.01 | 98.18±0.01 | 98.80±0.01 | 96.76±0.01 |
| 13. | ✓ | ✗ | ✓ | ✓ | 95.79±0.08 | 95.73±0.12 | 93.29±0.07 | 98.45±0.01 | 98.25±0.01 | 98.80±0.03 | 96.72±0.03 |
| 14. | ✗ | ✓ | ✓ | ✓ | 95.80±0.03 | 95.81±0.07 | 93.33±0.09 | 98.41±0.01 | 98.24±0.01 | 98.77±0.01 | 96.73±0.03 |
| 15. | ✓ | ✓ | ✓ | ✓ | 95.82±0.10 | 95.80±0.12 | 93.38±0.08 | 98.43±0.02 | 98.20±0.04 | 98.76±0.01 | 96.73±0.04 |

Table 8: Detailed results for probing tasks.

| | Pattern | | | | Encoders&Datasets | | | | | |
|---|---|---|---|---|---|---|---|---|---|---|
| | | | | | BERT-frozen | | | BERT-finetune | | |
| | (a) | (b) | (c) | (d) | ACE04 | ACE05 | Avg. | ACE04 | ACE05 | Avg. |
| i | Max pooling | | | | 75.54±0.43 | 76.90±0.31 | 76.22±0.12 | 84.12±0.09 | 82.56±0.12 | 83.34±0.06 |
| 1. | ✓ | ✗ | ✗ | ✗ | 79.17±0.19 | 79.38±0.32 | 79.28±0.14 | 83.92±0.13 | 82.94±0.02 | 83.43±0.07 |
| 2. | ✗ | ✓ | ✗ | ✗ | 80.00±0.28 | 80.01±0.30 | 80.01±0.11 | 84.23±0.11 | 83.50±0.12 | 83.86±0.10 |
| 3. | ✗ | ✗ | ✓ | ✗ | 80.61±0.27 | 79.97±0.12 | 80.29±0.10 | 84.38±0.20 | 83.52±0.26 | 83.95±0.10 |
| 4. | ✗ | ✗ | ✗ | ✓ | 77.39±0.24 | 78.97±0.30 | 78.18±0.16 | 83.71±0.15 | 83.31±0.30 | 83.51±0.21 |
| 5. | ✓ | ✓ | ✗ | ✗ | 80.37±0.05 | **80.57±0.29** | 80.47±0.14 | 84.33±0.16 | 83.35±0.26 | 83.84±0.21 |
| 6. | ✓ | ✗ | ✓ | ✗ | **81.31±0.17** | 80.36±0.32 | **80.83±0.23** | 84.54±0.28 | 83.63±0.38 | 84.08±0.33 |
| 7. | ✓ | ✗ | ✗ | ✓ | 77.69±0.29 | 78.58±0.08 | 78.14±0.18 | 84.54±0.20 | 82.72±0.24 | 83.63±0.18 |
| 8. | ✗ | ✓ | ✓ | ✗ | 79.65±0.20 | 79.67±0.07 | 79.66±0.13 | 84.21±0.15 | 83.20±0.30 | 83.70±0.19 |
| 9. | ✗ | ✓ | ✗ | ✓ | 79.54±0.25 | 79.90±0.25 | 79.72±0.21 | 84.25±0.13 | 83.60±0.33 | 83.93±0.21 |
| 10. | ✗ | ✗ | ✓ | ✓ | 79.49±0.33 | 79.09±0.20 | 79.29±0.16 | 83.83±0.28 | 83.17±0.32 | 83.50±0.28 |
| 11. | ✓ | ✓ | ✓ | ✗ | 80.23±0.33 | 80.07±0.29 | 80.15±0.22 | 84.57±0.35 | 83.54±0.09 | 84.05±0.13 |
| 12. | ✓ | ✓ | ✗ | ✓ | 79.49±0.16 | 80.23±0.23 | 79.86±0.09 | 84.32±0.14 | 83.49±0.34 | 83.90±0.19 |
| 13. | ✓ | ✗ | ✓ | ✓ | 79.14±0.31 | 79.11±0.26 | 79.12±0.09 | 84.58±0.06 | **83.70±0.24** | 84.14±0.12 |
| 14. | ✗ | ✓ | ✓ | ✓ | 79.66±0.16 | 80.38±0.24 | 80.02±0.10 | **84.97±0.19** | 83.56±0.25 | **84.26±0.04** |
| 15. | ✓ | ✓ | ✓ | ✓ | 78.86±0.24 | 79.80±0.30 | 79.33±0.05 | 84.36±0.04 | 83.50±0.31 | 83.93±0.16 |

Table 9: Detailed results for nested NER.

| | Pattern | | | | Encoders&Datasets | | | | | |
| | (a) | (b) | (c) | (d) | SpanBERT-frozen | | | SpanBERT-finetune | | |
| | | | | | REF | SRC | Avg. | REF | SRC | Avg. |
|---|---|---|---|---|---|---|---|---|---|---|
| i | Max pooling | | | | 95.62±0.06 | 92.88±0.06 | 94.25±0.06 | 96.52±0.21 | 93.66±0.01 | 95.09±0.11 |
| 1. | ✓ | ✗ | ✗ | ✗ | 95.90±0.02 | 93.40±0.09 | 94.65±0.05 | 96.65±0.05 | 93.77±0.03 | 95.21±0.04 |
| 2. | ✗ | ✓ | ✗ | ✗ | 95.83±0.05 | 93.39±0.10 | 94.61±0.07 | 96.67±0.09 | 93.86±0.01 | 95.27±0.05 |
| 3. | ✗ | ✗ | ✓ | ✗ | 95.73±0.01 | 93.43±0.04 | 94.58±0.02 | 96.52±0.18 | 93.81±0.01 | 95.17±0.09 |
| 4. | ✗ | ✗ | ✗ | ✓ | 95.93±0.07 | 93.22±0.03 | 94.58±0.05 | 96.73±0.16 | 93.77±0.02 | 95.25±0.09 |
| 5. | ✓ | ✓ | ✗ | ✗ | 95.86±0.05 | 93.45±0.01 | 94.66±0.02 | 96.73±0.07 | 93.78±0.01 | 95.26±0.04 |
| 6. | ✓ | ✗ | ✓ | ✗ | 95.87±0.03 | **93.51±0.01** | **94.69±0.02** | 96.71±0.24 | **93.87±0.01** | 95.29±0.12 |
| 7. | ✓ | ✗ | ✗ | ✓ | **96.01±0.15** | 93.12±0.05 | 94.57±0.10 | 96.62±0.12 | 93.80±0.09 | 95.21±0.10 |
| 8. | ✗ | ✓ | ✓ | ✗ | 95.62±0.05 | 93.47±0.01 | 94.55±0.03 | 96.59±0.30 | 93.84±0.01 | 95.22±0.15 |
| 9. | ✗ | ✓ | ✗ | ✓ | 95.94±0.06 | 93.25±0.07 | 94.60±0.07 | **96.83±0.17** | 93.79±0.08 | 95.31±0.12 |
| 10. | ✗ | ✗ | ✓ | ✓ | 95.92±0.08 | 93.19±0.03 | 94.56±0.06 | 96.69±0.23 | 93.81±0.05 | 95.25±0.14 |
| 11. | ✓ | ✓ | ✓ | ✗ | 95.97±0.15 | 93.41±0.13 | **94.69±0.14** | 96.60±0.17 | 93.85±0.01 | 95.23±0.09 |
| 12. | ✓ | ✓ | ✗ | ✓ | 96.00±0.09 | 93.28±0.02 | 94.64±0.06 | 96.74±0.20 | 93.78±0.04 | 95.26±0.12 |
| 13. | ✓ | ✗ | ✓ | ✓ | 95.93±0.09 | 93.21±0.06 | 94.57±0.08 | 96.80±0.08 | **93.87±0.02** | **95.34±0.05** |
| 14. | ✗ | ✓ | ✓ | ✓ | 95.78±0.09 | 93.16±0.08 | 94.47±0.09 | 96.80±0.10 | 93.84±0.01 | 95.32±0.05 |
| 15. | ✓ | ✓ | ✓ | ✓ | 95.87±0.10 | 93.25±0.04 | 94.56±0.07 | 96.59±0.01 | 93.75±0.04 | 95.17±0.02 |

Table 10: Detailed results for REF and SRC when SpanBERT is used as backbone.