# OpenReview forum: "Improving Span Representation by Efficient Span-Level Attention"
_EMNLP/2023/Conference — EMNLP 2023 Findings_

### Official Review · Reviewer_Qvm2 · 2023-07-20

**Soundness:** 3

**Excitement:**

2: Mediocre: This paper makes marginal contributions (vs non-contemporaneous work), so I would rather not see it in the conference.

**Paper Topic And Main Contributions:**

This paper studies how to perform attention for span representations, and designs 4 attention patterns to reduce the complexity. Experiments on a number of tasks show the effectiveness of the approach.

**Questions For The Authors:**

A, how about using 4 token-level encoder layers on top of BERT?

B, is it possible to automatically aggregate the 4 attention patterns?

thanks for the response, I am satisfied with the response for question A.

**Reasons To Accept:**

The paper presents 4 effective attention patterns for the computation of span representations, and experiment results show the effectiveness of the approach.

**Reasons To Reject:**

A, the improvements are not large.

B, the method has additional costs (4 more encoder layers at span-level in addition to BERT), and adding token-level encoder layers on top of BERT may also improve the performance.

C, the paper fails to find a general solution for related problems, using containment, adjacency, and all-token attention patterns at the same time leads to the best performance on average for NER, while using inside-token and containment attention patterns leads to the best performance on average for the other tasks.

D, the paper lacks in-depth analysis for the co-relation between the attention patterns and specific tasks.

E, Table 3 might be wrong, the average F1 score improvements are not consistent with Table 1.

thanks for the response, B and E are well addressed.

**Reproducibility:**

3: Could reproduce the results with some difficulty. The settings of parameters are underspecified or subjectively determined; the training/evaluation data are not widely available.

**Reviewer Confidence:**

3: Pretty sure, but there's a chance I missed something. Although I have a good feel for this area in general, I did not carefully check the paper's details, e.g., the math, experimental design, or novelty.

---

> ### Author Rebuttal · Authors · 2023-08-29
>
> We thank the reviewer for the detailed reviews and suggestions. It helps us improve the quality of our work.
>
> ## Reasons To Reject
>
> ### Reason A
>
> >the improvements are not large.
>
> Probing tasks are relatively simple, which is why simple baselines like max-pooling can yield impressive results. When facing harder tasks such as nested NER, the performance improvement of our method over simple baselines becomes more pronounced. We will consider conducting more experiments on more complex span prediction tasks such as constituency parsing, relation extraction, etc, where the improvements are expected to be more significant.
>
> ### Reason B (also Question A in Questions For The Authors)
>
> >the method has additional costs (4 more encoder layers at span-level in addition to BERT), and adding token-level encoder layers on top of BERT may also improve the performance.
>
> We evaluate the performance when 4 more token-level encoder layers are used. We will add it into Table 1 as a new baseline. From the table below, it can be seen that employing 4 more token-level layers does lead to performance gains, but the degree of improvement is consistently smaller compared to our method.
>
> |                        |NEL|REF|SRC| CTL   |MED|CTD|
> |:----|:----|:----|:----|:----|:----|:----|
> |           Max pooling | 95.61 | 95.58 | 92.85 | 98.00 | 97.98 | 98.19 |
> |   4 token-level layers | 95.73 | 95.76 | 93.32 | 98.34 | 98.04 | 98.61 |
> | Best attention pattern | 96.08 | 96.06 | 93.62 | 98.50 | 98.28 | 98.83 |
>
> ### Reason C
>
> >the paper fails to find a general solution for related problems, using containment, adjacency, and all-token attention patterns at the same time leads to the best performance on average for NER, while using inside-token and containment attention patterns leads to the best performance on average for the other tasks.
>
> It is natural that the performance of different attention patterns varies on different tasks and thus a general solution might not be possible.  In our paper, we have discussed principles to choose attention patterns for different tasks.
>
> * Lines 191 to 209: We categorize tasks into semantic and structural types, analyzing the rationales behind the corresponding optimal attention patterns. For tasks that place more emphasis on structures, attention patterns inspired by structural span interactions (Containment, Adjacency) show better performance. For tasks that prioritize textual content, the Inside-Token and All-Token attention patterns perform better due to their attention to the origin input text. This observation can serve as a guidence for combining attention patterns when encountering tasks of different types.
> * Lines 212 to 217: We suggest a reliable choice of attention pattern combination: Inside-Token and Containment.
> * Lines 229 to 233: We verify that the combination of Inside-Token and Containment/Adjacency performs well for nested NER when frozen BERT is used as the encoder backbone, aligning with experimental results for probing tasks under similar settings.
> * Lines 233 to 238: We provide additional advice that All-Token should be included in combination when using a stronger encoder backbone.
>
> ### Reason D
>
> >the paper lacks in-depth analysis for the co-relation between the attention patterns and specific tasks.
>
> In Lines 191 to 217, we have discussed the relations between tasks and attention patterns. Given the limited space of a short paper, we find it challenging to conduct more in-depth analysis.
>
> ### Reason E
>
> >Table 3 might be wrong, the average F1 score improvements are not consistent with Table 1.
>
> Table 3 is indeed accurate and consistent with Table 1. Note that the corresponding experiments of Table 3 were conducted on three tasks: REF, SRC and CTL, as stated in Lines 241 to 242.
>
> ## Questions For The Authors
>
> ### Question B
>
> >is it possible to automatically aggregate the 4 attention patterns?
>
> As we discussed in Lines 265 to 271 in the Limitations section, an automated attention pattern combiner based on reinforcement learning may help choose the optimal pattern combination when encountering new tasks.

---

### Official Review · Reviewer_T2H8 · 2023-08-02

**Typos Grammar Style And Presentation Improvements:** 1. The authors do not explicitly stat…
**Soundness:** 3

**Excitement:**

3: Ambivalent: It has merits (e.g., it reports state-of-the-art results, the idea is nice), but there are key weaknesses (e.g., it describes incremental work), and it can significantly benefit from another round of revision. However, I won't object to accepting it if my co-reviewers champion it.

**Paper Topic And Main Contributions:**

This paper proposes a new span-level attention mechanism to enhance the span representation based on a powerful pre-trained language model. It focuses on 4 patterns: inside-token, containment, adjacency and all-token when designing the span-level attention. Experimental results on several span-level tasks show the effectiveness of this method, especially for nested NER tasks.

**Questions For The Authors:**

1. It seems that the “inside-token” pattern is a subset of the “all-token” pattern. Then what is the difference between using only “all-token” and using both of them? If they are equivalent, then item 4 and 7 in Table 1 should be the same.

2. Why only REF, SRC and CTL tasks are chosen for layer num analysis in P3.4, rather than including all 6 tasks in Table 1?

3. I think the exact meaning of rel() formula in P2 Method should be clarified. Moreover, the meaning of figure 1 may need extra explanation. For first sight, it’s hard to understand how spans are indicated in this diagram.


**Reasons To Accept:**

1.	This paper is well written and easy to follow.

2.	The proposed method shows superior performance over the baselines, especially on the nested NER tasks.


**Reasons To Reject:**

1.	This paper shows little novelty, as it is only a marginal improvement of the conventional attention mechanism based on some span-level inductive bias.

2.	The performance improvement over simple baselines like max-pooling is also marginal for probing tasks.

3.	Although 4 patterns are proposed in this paper, it seems hard to find a unified solution or guideline for how to combine them in different tasks. If we have to try all possible combinations every time, the practicality of this method would be significantly reduced.

4. The authors validate the effectiveness of the proposed span-level attention only on the backbone of BERT. The paper lacks experiments on other encoder backbones to further demonstrate the generality of the proposed method.

5. The authors do not compare their methods with other state-of-the-art methods for span-related tasks, such as SpanBERT, thus lacking some credibility.

6. The writing can be improved. There are some typos and unclear descriptions. Please refer to comments for detail.


**Reproducibility:**

3: Could reproduce the results with some difficulty. The settings of parameters are underspecified or subjectively determined; the training/evaluation data are not widely available.

**Reviewer Confidence:**

5: Positive that my evaluation is correct. I read the paper very carefully and I am very familiar with related work.

---

> ### Author Rebuttal · Authors · 2023-08-29
>
> We thank the reviewer for the detailed reviews and suggestions, and many thanks for pointing out the writing typos. We will fix them.
>
> ## Reasons To Reject
>
> ### Reason 1
>
> >This paper shows little novelty, as it is only a marginal improvement of the conventional attention mechanism based on some span-level inductive bias.
>
> The novelty of our work lies in both the task and the method.
>
> **Task:** The majority of current research is centered around improving representations for fine-grained tokens or coarse-grained sentences and documents. However, spans, which are equally important components of natural language, are often overlooked. We attempt to enhance the quality of span representations by doing span-level attention and capturing span-level interactions, which is currently under-explored.
>
> **Method:** Compared with previous work of span representation, our work has the following novelties: (i) it does not rely on external information such as labels and parse trees. (ii) It explicitly models span interactions via span-level attention. (iii) We design four span-level attention patterns that ensure both effectiveness and efficiency.
>
> ### Reason 2
>
> >The performance improvement over simple baselines like max-pooling is also marginal for probing tasks.
>
> Probing tasks are relatively simple, which is why simple baselines like max-pooling can yield impressive results. When facing harder tasks such as nested NER, the performance improvement of our method over simple baselines becomes more pronounced. We will consider conducting more experiments on more complex span prediction tasks such as constituency parsing, relation extraction, etc, where the improvements are expected to be more significant.
>
> ### Reason 3
>
> >Although 4 patterns are proposed in this paper, it seems hard to find a unified solution or guideline for how to combine them in different tasks. If we have to try all possible combinations every time, the practicality of this method would be significantly reduced.
>
> It is natural that the performance of different attention patterns varies on different tasks and thus a unified solution might not be possible.  In our paper, we have discussed principles to choose attention patterns for different tasks.
>
> * Lines 191 to 209: We categorize tasks into semantic and structural types, analyzing the rationales behind the corresponding optimal attention patterns. For tasks that place more emphasis on structures, attention patterns inspired by structural span interactions (Containment, Adjacency) show better performance. For tasks that prioritize textual content, the Inside-Token and All-Token attention patterns perform better due to their attention to the origin input text. This observation can serve as a guidence for combining attention patterns when encountering tasks of different types.
> * Lines 212 to 217: We suggest a reliable choice of attention pattern combination: Inside-Token and Containment.
> * Lines 233 to 238: We provide additional advice that All-Token should be included in combination when using a stronger encoder backbone.
>
> ### Reason 4 and 5
>
> >4. The authors validate the effectiveness of the proposed span-level attention only on the backbone of BERT. The paper lacks experiments on other encoder backbones to further demonstrate the generality of the proposed method.
> >5. The authors do not compare their methods with other state-of-the-art methods for span-related tasks, such as SpanBERT, thus lacking some credibthe ility.
>
> Below we provide the results of experiments when utilizing SpanBERT as the backbone. Due to limited rebuttal time, we only conduct experiments on the REF and SRC tasks. The results of frozen SpanBERT are presented in the first table, while those of fine-tuned SpanBERT are presented in the second table. We use the same index as Table 1 in our paper in column 1 and 4.  As we can see, our proposed method still brings performance gain after changing the backbone to SpanBERT, no matter fine-tuned or not. It further demonstrates the generality of our method (in response to reason 4).
>
> Moreover, Fine-tuned SpanBERT with max pooling is a widely favored choice for span-related tasks, and our results show that applying span-level attention on this backbone can still bring performance improvement, which means our proposed method outperforms "other state-of-the-art methods" (in response to reason 5).
>
> |Frozen SpanBERT|REF|SRC|    |REF|SRC|
> |:----|:----|:----|:----|:----|:----|
> | Max pooling |95.62|92.88|  8 |95.62|93.47|
> |           1 |95.90|93.40|  9 |95.94|93.25|
> |           2 |95.83|93.39| 10 |95.92|93.19|
> |           3 |95.73|93.43| 11 |95.97|93.41|
> |           4 |95.93|93.22| 12 |96.00|93.28|
> |           5 |95.86|93.45| 13 |95.93|93.21|
> |           6 |95.87|**93.51**| 14 |95.78|93.16|
> |           7 |**96.01**|93.12| 15 |95.87|93.25|
>
> |Fine-tuned SpanBERT|REF|SRC|    |REF|SRC|
> |:----|:----|:----|:----|:----|:----|
> | Max pooling |96.52|93.66|  8 |96.59|93.84|
> |           1 |96.65|93.77|  9 |**96.83**|93.79|
> |           2 |96.67|93.86| 10 |96.69|93.81|
> |           3 |96.52|93.81| 11 |96.60|93.85|
> |           4 |96.73|93.77| 12 |96.74|93.78|
> |           5 |96.73|93.78| 13 |96.80|**93.87**|
> |           6 |96.71|**93.87**| 14 |96.80|93.84|
> |           7 |96.62|93.80| 15 |96.59|93.75|
>
> ### Reason 6
>
> >The writing can be improved. There are some typos and unclear descriptions. Please refer to comments for detail.
>
> Thanks for pointing them out. We will fix them.
>
> ## Questions For The Authors
>
> ### Question 1
>
> >It seems that the “inside-token” pattern is a subset of the “all-token” pattern. Then what is the difference between using only “all-token” and using both of them? If they are equivalent, then item 4 and 7 in Table 1 should be the same.
>
> When Inside-Token and All-Token are combined, tokens inside a span is attended to twice by the span, instead of once when only All-Token is used.
>
> ### Question 2
>
> >Why only REF, SRC and CTL tasks are chosen for layer num analysis in P3.4, rather than including all 6 tasks in Table 1?
>
> Due to limitations of computational resources, we only choose these three tasks for layer num analysis, which cover both semantic and sturctural tasks, as stated in Lines 242 to 243. The other three tasks are named entity labeling (NEL), mention detection (MED) and constituent detection (CTD). MED is a complement for coreference arc prediction (REF) and shares similar task properities, analogous to how CTD complements constituent labeling (CTL). Therefore, these two tasks are not selected. NEL is not chosen due to its higher performance variance.
>
> ### Question 3
>
> >I think the exact meaning of rel() formula in P2 Method should be clarified. Moreover, the meaning of figure 1 may need extra explanation. For first sight, it’s hard to understand how spans are indicated in this diagram.
>
> Thanks for the suggestions. We will incorporate additional explanations for formulas and figures if the paper is accepted and an extra page is given.
>
> ## Typos Grammar Style And Presentation Improvements
>
> ### Improvement 1
>
> >1. The authors do not explicitly state the intuitions of the pattern setting. The authors do not explore other possible patterns or combinations, such as overlapping spans, non-contiguous spans, etc.
>
> Thanks for the suggention. We have shown the intuitive advantages of each pattern in Lines 116 to 137, which is also part of our intuitions. We will modify this paragraph to present our intuitions more explicitly. We deliberately avoid patterns such as overlapping spans and non-contiguous spans, because considering these patterns will increase the time complexity to at least $O(n^4)$, which is not practical. We only consider patterns with cubic time complexity for the sake of efficiency.
>
> ### Improvement 3-5
>
> >3. As the major goal of this paper is to improve span representation, related works such as [1] [2] conduct experiments on NLU and QA tasks, which are also span-related tasks. Experimental results extended on those tasks would be helpful.
> >4. There are some missing related works such as [1] [2], it would be better to add more relevant discussions.
> >5. Minor comments on writing:
> >(1) Line #039: lies on -> lies in
> >(2) Line #072: Comparing to -> Compared to
>
> Thanks for pointing out the missing references and writing typos. We will include more tasks in subsequent updates. We do not see what reference [1] and [2] are in your review. Could you please provide them?

---

### Official Review · Reviewer_iZpj · 2023-08-05

**Soundness:** 4

**Excitement:**

3: Ambivalent: It has merits (e.g., it reports state-of-the-art results, the idea is nice), but there are key weaknesses (e.g., it describes incremental work), and it can significantly benefit from another round of revision. However, I won't object to accepting it if my co-reviewers champion it.

**Missing References:**

1. Improving Span Representation for Domain-adapted Coreference Resolution https://aclanthology.org/2021.crac-1.13.pdf
2. A Structured Span Selector https://aclanthology.org/2022.naacl-main.189/


**Paper Topic And Main Contributions:**

This paper studies the effect of 4 different span-level attention pruning schemes on downstream span classification tasks. This paper proposes 4 heuristics for span pruning to reduce the $O(n^4)$ complexity of span-level attention to $O(n^3)$. Extensive experiments are conducted on a variety of span classification tasks to show the efficacy of the proposed heuristics in each task.

**Questions For The Authors:**

A. The mathematical formula of span representations should be provided in the paper even if plain transformers are used.

B. How do the pruning heuristics proposed in the paper perform compared with the simple span pruning method by Lee et al., 2017? I.e., scoring all the spans and taking the top-k, where k is $O(n)$.

**Reasons To Accept:**

1. The experiments are extensive and solid.
2. The proposed heuristics are simple and useful.

**Reasons To Reject:**

I’m not sure how many meaningful spans (i.e., spans that are grammatical constituents, see reference 2) can be recalled by the proposed heuristics. It seems the majority of the spans recalled by heuristics (b) and (c) are not syntactic constituents. More discussions on this are needed. I.e., why is it helpful to consider fragment spans that are not syntactic constituents?

**Reproducibility:**

5: Could easily reproduce the results.

**Reviewer Confidence:**

5: Positive that my evaluation is correct. I read the paper very carefully and I am very familiar with related work.

---

> ### Author Rebuttal · Authors · 2023-08-29
>
> We thank the reviewer for the detailed reviews and suggestions, and many thanks for pointing out the missing references. We will add them.
>
> ## Reasons To Reject
>
> >I'm not sure how many meaningful spans (i.e., spans that are grammatical constituents, see reference 2) can be recalled by the proposed heuristics. It seems the majority of the spans recalled by heuristics (b) and (c) are not syntactic constituents. More discussions on this are needed. I.e., why is it helpful to consider fragment spans that are not syntactic constituents?
>
> First of all, we would like to argue that there is no consensus that distituents (i.e., spans that are not constituents) cannot provide useful information. For example, the model may learn distituent representations and propagate them via span-level self-attention to facilitate distinction between distituents vs. constituents.
>
> Second, there is also no guarantee that grammar-based approaches such as [2] do not output distituents. In fact, since they only output $2n-1$ spans from the predicted optimal parse tree, their recall of constituents can be low if the predicted parse does not align well with the ground-truth parse tree. Our approach, on the other hand, considers a much larger pool of spans and hence is very likely to have a higher recall of constituents. Span-level self-attention is then used to automatically identify useful spans (such as constituents) within the pool. Moreover, our heuristics prohibit crossing spans and thus make it less likely that a constituent span attends to a distituent span.
>
> ## Questions For The Authors
>
> ### Question A
>
> >The mathematical formula of span representations should be provided in the paper even if plain transformers are used.
>
> Thanks for the suggestion. We do not provide detailed formulas because of limited space of a short paper, but we will add them if the paper is accepted and an extra page is given.
>
> ### Question B
>
> >How do the pruning heuristics proposed in the paper perform compared with the simple span pruning method by Lee et al., 2017? I.e., scoring all the spans and taking the top-k, where k is $O(n)$
>
> Compared to the pruning method proposed by Lee et al. 2017, our method is more straightforward as it doesn't need an external scoring function. Additionally, our method does not involve discrete pruning decisions and hence circumvents the challenge of backpropagation through hard decisions.

---

### Meta-Review · Area_Chair_jj2A · 2023-09-19

**Recommendation:** 3

**Metareview:**

This paper proposes and analyzes pruning techniques to efficiently calculate span-level representations. Reviewers have found the proposed approach solid, and the experiments are thorough, whereas the novelty of this paper is somewhat limited.

---

### Decision · Program_Chairs · 2023-10-07

**Decision:**

Accept-Findings

**Comment:**

This paper proposes and analyzes pruning techniques to efficiently calculate span-level representations. Reviewers have found the proposed approach solid, and the experiments are thorough, whereas the novelty of this paper is somewhat limited.